# Dose Optimization in Oncology Drug Development: The Emerging Role of Pharmacogenomics, Pharmacokinetics, and Pharmacodynamics

**DOI:** 10.3390/cancers15123233

**Published:** 2023-06-18

**Authors:** Apostolos Papachristos, Jai Patel, Maria Vasileiou, George P. Patrinos

**Affiliations:** 1Regeneron Pharmaceuticals, Inc., Tarrytown, NY 10591, USA; 2Department of Cancer Pharmacology and Pharmacogenomics, Levine Cancer Institute, Atrium Health, Charlotte, NC 28204, USA; jai.patel@atriumhealth.org; 3Department of Pharmacy, School of Health Sciences, National and Kapodistrian University of Athens, 16121 Athens, Greece; mariavasileiou65@gmail.com; 4Laboratory of Pharmacogenomics and Individualized Therapy, Department of Pharmacy, School of Health Sciences, University of Patras, 26504 Patras, Greece; gpatrinos@upatras.gr; 5Department of Genetics and Genomics, College of Medicine and Health Sciences, United Arab Emirates University, Al Ain P.O. Box 15551, United Arab Emirates; 6Zayed Center for Health Sciences, United Arab Emirates University, Al Ain P.O. Box 15551, United Arab Emirates

**Keywords:** dose optimization, clinical trials, TKIs, ADCs, TDM, monoclonal antibodies, FDA, pharmacogenomics, pharmacodynamics, pharmacokinetics

## Abstract

**Simple Summary:**

Accelerated clinical development of anticancer drugs is crucial to ensure patients’ access to the most safe and effective treatments. At the same time, the development process should be designed to ensure that the optimal dose and schedule are administrated. Therefore, pharmacological methods such as pharmacogenomics, pharmacokinetics, and pharmacodynamics must be integrated to inform oncology drug development and dose optimization. Herein we present a summary and some examples of the utility of those methods.

**Abstract:**

Drugs’ safety and effectiveness are evaluated in randomized, dose-ranging trials in most therapeutic areas. However, this is only sometimes feasible in oncology, and dose-ranging studies are mainly limited to Phase 1 clinical trials. Moreover, although new treatment modalities (e.g., small molecule targeted therapies, biologics, and antibody-drug conjugates) present different characteristics compared to cytotoxic agents (e.g., target saturation limits, wider therapeutic index, fewer off-target side effects), in most cases, the design of Phase 1 studies and the dose selection is still based on the Maximum Tolerated Dose (MTD) approach used for the development of cytotoxic agents. Therefore, the dose was not optimized in some cases and was modified post-marketing (e.g., ceritinib, dasatinib, niraparib, ponatinib, cabazitaxel, and gemtuzumab-ozogamicin). The FDA recognized the drawbacks of this approach and, in 2021, launched Project Optimus, which provides the framework and guidance for dose optimization during the clinical development stages of anticancer agents. Since dose optimization is crucial in clinical development, especially of targeted therapies, it is necessary to identify the role of pharmacological tools such as pharmacogenomics, therapeutic drug monitoring, and pharmacodynamics, which could be integrated into all phases of drug development and support dose optimization, as well as the chances of positive clinical outcomes.

## 1. Introduction

Pharmacogenomics aims to decipher the role of genetic variants on drug efficacy and toxicity. The term “pharmacogenetics” was first introduced by Frederich Vogel, a German geneticist, in 1959 [1]. The first applied genotyping study was conducted in the 1970s and involved the altered metabolism of antihypertensive debrisoquine and antiarrhythmic sparteine [2,3]. Years later, the altered metabolism was attributed to the genetic variants of the cytochrome P450 2D6 (*CYP2D6*) gene [4]. Among the most commonly studied genomic variations are germline variants, epigenetic variants, variations in gene expression profiles, and structural changes in chromosomes. Such genetic variations can be inherited (germline) or acquired (somatic) [5].

Clinical evidence shows significant variability in the response to pharmacological agents between patients (inter-individual variability). While many variables have been associated with drug response (gender, diet, renal and hepatic function, pregnancy, drug-drug interactions, and drug-food interactions), the genetic profile can have a major impact on treatment outcomes [6]. A greater understanding of the inter-individual genetic variability in a population has the potential to revolutionize the use of many medications. This is particularly important in the field of oncology since cancer is a leading cause of morbidity, and failed or toxic treatment can be life-threatening [7]. Administration of anticancer drugs has been linked to toxicity, ranging from mild to lethal adverse events. A well-established example in the case of breast, colon, and gastrointestinal cancer is the standard treatment with antimetabolite fluorouracil. Monotherapy with fluorouracil has been associated with grade 3/4 toxicity (diarrhea, mucositis, leucopenia) in 16% of patients [8]; however, when used in combination regimens, as typically observed in most gastrointestinal malignancies, the rate of grade 3/4 toxicities increases to 30–35%. Given that 80–85% of fluorouracil is metabolized by the rate-limiting enzyme dihydropyrimidine dehydrogenase (*DPYD*), a loss or reduction of enzyme activity could potentially lead to significantly prolonged half-life (up to 100-fold) resulting in the prolonged cytotoxic effect of the drug and significant toxicity [9]. Similarly, genetic variations play an essential role in the pharmacokinetics of irinotecan, a camptothecin derivative used to treat several types of solid tumors. Uridine diphosphate glucuronosyltransferase 1A1 (*UGT1A1*) conjugates the active metabolite of irinotecan, SN-38, to an inactive glucuronide metabolite, SN-38G, for ultimate excretion from the body. More than 60 genomic variants in the *UGT1A1* gene have been identified. Of these, the most common and studied variant in the Caucasian population is the *28 allele, which results in up to a 70% reduction in gene expression and an increased risk of irinotecan-induced toxicity [9]. Hence, it is evident that severe adverse events can be mitigated through the implementation of pharmacogenomic testing for dose optimization.

In the era of personalized medicine, optimizing treatment strategy according to the molecular and genetic profile of a patient is an ideal scenario. Various legislative and regulatory bodies are actively developing policies regarding genome-guided treatment in drug labels. The European Medicines Agency (EMA) recommends testing for *DPYD* deficiency prior to the initiation of any fluoropyrimidine. In fact, patients with complete *DPYD* deficiency, also termed as poor metabolizers, should not be administered fluorouracil derivatives, while partially deficient patients (intermediate metabolizers) should receive lower doses [10]. The US Food and Drug Administration (FDA) also includes germline pharmacogenomic information in the drug label for belinostat, belzutifan, binimetinib, capecitabine, cisplatin, dabrafenib, erdafitinib, fluorouracil, gefitinib, irinotecan, lenalidomide, mercaptopurine, nilotinib, pazopanib, rucaparib, sacituzumab-govitecan, tamoxifen, thioguanine, and trametinib [11]. Despite the strong evidence and inclusion of pharmacogenomic information in labels, multiple barriers impede the widespread adoption of germline pharmacogenomic testing for the selection of the optimal dose in clinical practice and in drug development.

Pharmacogenomics (PGx), pharmacokinetics (PK), and pharmacodynamics (PD) are the primary pharmacological disciplines that could be utilized in each step of drug development to inform decisions. At the discovery and preclinical development stages, PK and PD are crucial to define and select the candidate molecules to be tested in the clinic. PGx could also be informative to identify specific targets and patient populations. At the early clinical development phase, PGx-guided, PK-guided (i.e., therapeutic drug monitoring (TDM)), and PD-guided approaches could be implemented for a dose selection of Phase 2 and 3 studies. In later clinical development phases, PGx and PK/PD could provide robust evidence for exposure-response relationships, dose optimization, and support regulatory submissions and approvals. Finally, post-marketing PGx and PD in Phase 4 studies could provide information on the drug’s effect in patients with rare genotypes or identify additional predictive biomarkers to inform dose adjustments, dose, and treatment optimization (Figure 1).

Representatives from the FDA recently published a review highlighting strategies to integrate dose optimization into premarketing drug development [12]. For example, identification of several candidate dosages and dosage ranges, determining whether there are PD biomarkers that could inform dose optimization, considering integrating model simulations with emerging clinical data to support dose optimization, conducting randomized dose trials, integrating safety beyond dose-limiting toxicities, collecting exposure-response data early, and refining dose optimization throughout clinical development.

As a result, the FDA’s Oncology Center of Excellence recently announced a new initiative, Project Optimus, focused on reforming dose optimization and selection in oncology drug development. The goal is to establish a dose-finding and optimization paradigm across oncology that emphasizes maximizing efficacy while improving safety and tolerability. In order to optimize the dose to achieve this goal, it is necessary to collect adequate data to characterize exposure-response (E-R) and well-describe the relationship between drug concentrations, efficacy, and safety. Especially in the early clinical development of new therapeutic modalities, the adoption of more sophisticated and precise approaches compared to the standard 3 + 3 method could significantly increase the chances of selecting the optimal dose, achieving the best clinical outcomes, and subsequently accelerating the approval of new drugs without the requirement of post-marketing dose optimization studies. In the 3 + 3 design, three patients are enrolled into each dose cohort, and if there is no dose-limiting toxicity (DLT) observed, the enrollment proceeds to the next higher dose cohort [13]. However, this methodology neither provides robust evidence nor allows adequate clinical, PK, and PD data collection to select the optimal dose. Herein we present the current trends in the clinical development of anticancer agents, FDA guidance for dose optimization, and examples of the evidence for using pharmacogenomics, TDM, and pharmacodynamics in drug development (Figure 2).

## 2. Dose-Finding Trials Design—Project Optimus

Drugs are evaluated in pivotal, randomized, dose-ranging trials in most therapeutic areas. This study design provides robust data to understand and characterize the effect of dose on both efficacy and toxicity. On the other hand, in oncology, there is an urgent need for rapid access to novel, effective, life-changing treatments, and optimal dose selection before approval following a sequential Phase 1–3 trial paradigm may not be feasible [14]. Thus, dose-ranging studies are mainly limited to first-in humans (FIH) Phase 1 clinical trials. Furthermore, the design of oncology FIH focuses primarily on identifying the MTD via monitoring for DLTs during the first treatment cycle or the highest-tested dose (HTD) if no DLTs are observed. This approach is based on the hypothesis that higher exposure leads to an increased antitumor effect. Hence, the dose is determined by DLTs and is selected to maximize clinical benefit [14,15,16,17,18]. Then MTD/HTD is used in registrational trials to assess efficacy and safety in a larger population rather than optimize the dose. This approach was initially developed for cytotoxic agents for which the E-R relationship is always steep in terms of efficacy and toxicity. However, oncology treatment options have evolved from cytotoxic agents to molecularly targeted agents (e.g., small molecules tyrosine kinase inhibitors (TKIs), biologics, antibody-drug conjugates (ADCs)) or immune-oncology (IO) agents [17,19]. These new modalities often have target saturation limits below the MTD, wider therapeutic index, and present fewer off-target side effects compared to cytotoxic agents. The E-R relationship is not profound or even flat; most toxicities occur months after treatment initiation. However, the dose selection is still based on observed MTD/HTD early in treatment [19].

As a result, following the MTD/HTD approach is not ideal for defining the optimal dose and schedule. Some examples of drugs whose doses or schedules were modified post-marketing include ceritinib, dasatinib, niraparib, ponatinib, cabazitaxel, and gemtuzumab-ozogamicin [19]. The FDA has recognized the drawbacks of the MTD/HTD approach. In 2021, the FDA Oncology Center of Excellence launched Project Optimus. It released the draft guidance “Optimizing the Dosage of Human Prescription Drugs and Biological Products for the Treatment of Oncologic Diseases Guidance for Industry,” which recommends dose-ranging studies to evaluate the benefit-risk ratio at the early clinical development stages of a drug prior to pivotal studies. Specifically, the FDA recommends the design, when feasible, of randomized, parallel dose-response trials that will compare the activity and safety of multiple dosages. Initially, dosages should be selected based on relevant preclinical and clinical data, including robust pharmacological studies. Multiple dosage comparisons should be performed prior to registration trials or as part of a registrational trial by adding dosage arms [14,20].

The main barrier to determining the optimal dose is the large interpatient pharmacokinetic variability, which is the result of non-tumor-specific factors such as physiologic factors, patient characteristics, drug-drug interactions, and environmental factors [21]. Genetics also plays a crucial role, and genomic variants in drug-metabolizing enzymes and transporter proteins are a primary source of pharmacokinetic variability [7,21].

Several genetic variations affecting the pharmacokinetics of anticancer drugs have been identified and should be considered in clinical trial design. Suppose in a Phase 1 trial of a new agent subject to P450 metabolism, a subset of patients are poor metabolizers (PM). In that case, DLTs may be observed earlier in a group of patients genetically predisposed to toxicity, which will lead to sub-optimal dose selection for the general population and may also put the program on hold due to safety concerns that will delay the drug development process or may even put the development plan at risk. On the other hand, if, due to the small sample size, most patients in a Phase 1 study are normal (NM) or ultrarapid (UM) metabolizers, the selected dose for pivotal studies may not be tolerated in PMs at the larger Phase 2 and 3 trials. A good example of the implementation of pharmacogenomics for dose finding and optimization is the selective mesenchymal-epithelial transition factor (c-MET) inhibitor tivantinib metabolized by cytochrome P450 2C19 (*CYP2C19*). Since approximately 20% of Asians are *CYP2C19* poor metabolizers, all patients enrolled in the open-label, dose-finding Phase 1 study in Japan were prospectively tested for the *CYP2C19* genotype and divided into two groups. NMs carry two *CYP2C19**1 alleles, whereas PMs carry either two of the *CYP2C19**2 and/or *CYP2C19**3 alleles. Results showed that exposure was almost double in PMs, and tivantinib was well tolerated up to 360 mg twice daily for NMs and 240 mg twice daily for PMs, which was then carried forward as the recommended Phase 2 doses [22]. This paradigm of dose stratification based on genotype/phenotype allows for drug exposure and safety normalization, which inevitably improves tolerability and, thus, efficacy. Further, the inclusion of this approach during clinical development and eventual FDA approval will ensure upfront pharmacogenomics testing to guide dosing in the post-marketing setting in contrast to situations like *DPYD* and *UGT1A1* for fluoropyrimidines and irinotecan, respectively, which have not been routinely adopted in the U.S. Similarly, the evaluation of genotype information is recommended by the FDA for antibody-drug conjugates (ADCs), as specific genetic variants may affect the metabolism and transport of the unconjugated payload and subsequently affect the toxicity profile of the ADC [23].

A characteristic example requiring further post-marketing dose optimization studies is the recently approved KRAS inhibitor sotorasib. A Phase I trial evaluated sotorasib in 129 patients with Kirsten rat sarcoma viral oncogene homolog glycine 12 to cysteine (*KRAS* G12C)-mutated advanced solid tumors, including 59 with NSCLC [24]. The planned dose levels were 180 mg, 360 mg, 720 mg, and 960 mg. No dose-limiting toxic effects were observed, but more than half experienced any-grade treatment-related adverse events, with 12% experiencing grade 3 or 4 events. The greatest efficacy benefit was observed in those with NSCLC. A Phase 2 trial evaluated sotorasib 960 mg in 126 patients with *KRAS* G12C-positive NSCLC [25]. The results showed a 36% response rate, 81% disease control rate, 70% experienced any-grade adverse events, and 20% experienced a grade 3 or 4 event. However, there was no evidence of a dose-response relationship, raising the question of whether higher doses contributed to improved response or simply an increase in adverse events. Subsequently, the FDA mandated that the sponsor conduct a randomized trial comparing 960 mg daily with 240 mg daily, the results of which are currently pending. The Phase 3 study of sotorasib 960 mg versus docetaxel in previously treated NSCLC showed a modest 1.1-month improvement in PFS with one-third of patients experiencing grade 3 or higher treatment-related adverse events, again suggesting that the higher dose of 960 mg is likely not optimal [26].

Dose optimization is clearly crucial in the clinical development of anticancer agents and especially targeted therapies. In specific cases, pharmacogenomics, TDM, and pharmacodynamics could be valuable tools to integrate, especially in early clinical trials and dose-ranging studies. This individualized dosing approach will accelerate the drug development process, ensure dose optimization, and increase the chances of positive clinical outcomes. The following section provides some anticancer drug examples in which dose optimization using various pharmacological methods could improve drug safety and response.

## 3. Modeling Simulation (M&S) in Dose Selection

A crucial step to describe and predict the exposure over time and the effects resulting from a specific dosing regimen is PK/PD analyses and modeling [27]. Modeling and simulation (M&S) are powerful tools that complement traditional methods for gathering evidence and have rendered growing contributions to inform decision-making processes on drug clinical development. M&S integrates computer-aided mathematical simulation and biological sciences to combine pre-clinical, clinical, and historical published data to investigate the relationships between exposure and clinical outcomes. Interestingly, models could be developed to inform decisions in all steps of drug development, including pathway, target, and molecule identification, clinical trial design, and dose selection [28,29]. A study by Milligan and coworkers analyzed a pool of 68 failed Phase 2–4 clinical trials and found that M&S could have provided important information regarding the exposure-response relationship, mechanism of action, and treatment approach in specific populations that could have informed design and prevented failure [30].

M&S also plays a crucial role in regulatory approval, and pharmacometrics reviews are conducted during submission. The FDA, EMA, Pharmaceuticals and Medical Devices Agency (PMDA), and other key regulatory agencies have incorporated M&S in their review, and sponsors are expected to perform PK/PD M&S studies using clinical data and include them in the information package to support dose selection and labeling [31,32].

Finally, pharmacometrics modules and M&S studies have widely been used to analyze clinical trials data to inform first-in-human dose selection, recommended Phase 2 dose optimization, investigate and define complex issues such as the effect of drug-drug interactions (DDI), age, body weight, pregnancy, hepatic and renal function, and genetic variants on PK and PD of a drug. Therefore, M&S and pharmacometrics tools should be the backbone for the optimal design of PGx studies and data analysis, subsequently informing dose optimization decisions based on genetic variations. A characteristic example of the utility of M&S and pharmacometrics in this setting is the study by Minichmayr and coworkers. Authors combined two PK and one PD (myelosuppression) models to simulate and predict the exposure of irinotecan and its metabolite SN-38 for different genetic variants of *UGT1A1* and its association with neutropenia following conventional versus PGx-based irinotecan dosing (350 vs. 245 mg/m^2^). This study provides the framework for the successful implementation of M&S methods to inform the study design in order to generate data and provide robust evidence about any significant association between genetic variants, drug exposure, and clinical outcomes [33].

## 4. Tyrosine Kinase Inhibitors and Therapeutic Drug Monitoring to Optimize Dose/Exposure

Tyrosine kinase inhibitors (TKIs) are prescribed at a fixed starting dose, despite their high pharmacokinetic inter-patient variability. Multiple factors contribute to this variability, such as age, body mass index, compliance, concomitant drugs, diet, genetics, and organ function. Systemic drug exposure measured by area-under-the-curve (AUC) or steady-state concentrations is related to efficacy and toxicity for many TKIs [34]. Josephs and coworkers [35] and Yu and coworkers [34] previously described the clinical pharmacokinetics of TKIs and data on the exposure-response relationships and guidance on potential pharmacokinetic-guided dose modifications. Target concentrations or AUCs have been proposed to maximize clinical response and reduce toxicities for TKIs such as axitinib, crizotinib, dasatinib, erlotinib, gefitinib, ibrutinib, imatinib, lapatinib, nilotinib, pazopanib, sorafenib, sunitinib, trametinib, vandetanib, and vemurafenib, although none were tested during clinical development [36].

For example, a study of 110 patients with chronic myeloid leukemia (CML) explored the relationship between imatinib concentration and clinical response and adverse reactions, followed by the impact of dose reductions [37]. There was a significant difference in imatinib plasma concentrations between those who had a major molecular response compared to those not achieving a response (1473.70 ± 419.13 vs. 985.8 ± 213.32 ng/mL) when receiving 400 mg daily. Concentrations above 1000 ng/mL predicted improved event- and failure-free survival, whereas those above 1685 ng/mL predicted toxicities, including diarrhea and edema. One-third underwent dose reductions; those with higher concentrations after dose reduction were more likely to maintain a molecular response while avoiding adverse reactions. Similarly, investigators identified significant associations between axitinib exposure and efficacy in a pooled analysis from Phase 3 trials [38]. Data from 168 patients with metastatic renal cell carcinoma were available. The median AUC at the end of 4 weeks with 5 mg twice daily dosing was 375 ng·h/mL (range 32.8–1728 ng·h/mL). There was a 1.5-fold increase in the probability of achieving a partial response for every 100 ng·h/mL increase in AUC. Based on prior data, a threshold of 300 ng·h/mL was used to correlate with efficacy outcomes [39]. Median PFS and OS were significantly longer in those with AUC > 300 ng·h/mL (HR 0.871 and 0.810; *p* < 0.001 for both). In a subsequent study, investigators showed that those who achieved higher axitinib exposure had better clinical outcomes [40]. At the end of a 4-week lead-in, patients who were tolerating treatment were randomly assigned to either axitinib or placebo dose escalation. About half of all patients were candidates for dose escalation, which significantly improved drug exposure and response rates (54% vs. 34%; risk ratio 1.58, *p* = 0.019). These studies highlight examples in which TKIs demonstrate significant interpatient pharmacokinetic variability and the importance of maximizing exposure without compromising tolerability.

The ability to integrate pharmacokinetic-guided dosing in the clinical setting should also be evaluated. In a prospective multicenter study evaluating pharmacokinetic-guided dosing in 600 patients with 24 different oral targeted therapies (mostly TKIs), investigators demonstrated that the proportion of underexposed patients at 12 weeks was reduced by 39% compared with historical data [41]. At 12 weeks, 26% of evaluable patients had low exposure compared to 42% in historical data. In total, more than half had at least one pharmacokinetic sample below the preset target at a given time point during treatment, in which another half had pharmacokinetic-guided interventions. These interventions were dependent on the specific drug and resulted in about three-quarters of patients achieving the target concentration range. Overall, successful pharmacokinetic-guided interventions were implemented in 21% of patients, while nearly half of all patients had adequate exposure during the collection time points and did not require dosing interventions. This study showed that a large proportion of patients receiving oral targeted therapies (mostly TKIs) do not achieve target drug concentrations which could compromise efficacy and/or increase toxicity risk. While it was feasible to implement pharmacokinetic-guided dosing in most patients, this study did not evaluate clinical outcomes [41].

A review by Groenland and coworkers [42] identified seven criteria for drugs to be considered suitable candidates for TDM, including the absence of an easily measurable biomarker for drug effect, long-term therapy, availability of a validated, sensitive bioanalytical method to detect drug concentrations, interpatient variability in pharmacokinetic exposure, narrow therapeutic range, defined and consistent exposure-response relationships, and feasible dose-adaptation strategies. They further claim that these requirements are met for most oral targeted therapies, including imatinib, pazopanib, sunitinib, everolimus, and endoxifen. While most studies on dose optimization using TDM have been performed post-marketing, there is an immense opportunity to capitalize on the exposure-response relationship in early drug development. It is critical that newer targeted therapies undergo robust PK and PD studies to understand the inter-patient variability at different dose levels and the relationship between exposure and response. This may not only inform the potential Phase 2 recommended dose(s) but also provide guidance on whether TDM might be useful to target individualized doses on a patient-specific level [12]. It may also help determine what biological and clinical factors impact drug concentrations.

## 5. Tyrosine Kinase Inhibitors and Therapeutic Drug Monitoring to Optimize Dose/Exposure

Examples such as *DPYD*, nudix hydrolase 15 (*NUDT15*), thiopurine methyltransferase (*TPMT*), and *UGT1A1* are well established. These drug-gene associations are mentioned in the FDA Table of Pharmacogenetic Associations under associations for which the data support therapeutic management, and guidelines from the Clinical Pharmacogenetics Implementation Consortium exist for *CYP2D6*, *DPYD*, *NUDT15*, and *TPMT*, underscoring their importance in optimizing dose selection to reduce drug toxicity and/or improve efficacy. In keeping with the theme of emerging evidence with TKIs, this section highlights the potential role of pharmacogenomics with TKIs. Given that many TKIs are hepatically metabolized by cytochrome P450s and undergo drug efflux via p-glycoprotein, it is conceivable that pharmacogenetic variation in drug metabolism genes and/or transporters may alter drug exposure and response. A systematic review of PGx and TKIs in renal cell carcinoma from 2004 to 2015 identified 54 studies that evaluated the effect of SNPs on various efficacy and toxicity endpoints [43]. An increased risk of sunitinib-related toxicities was noted in patients with SNPs in cytochrome P450 1A1 (*CYP1A1*), nuclear receptor subfamily 1 group I member 3 (*NR1I3*), adenosine 5′-triphosphate–binding cassette subfamily B member 1 (*ABCB1*), and adenosine 5′-triphosphate–binding cassette subfamily G member 2 (*ABCG2*). The cytochrome P450 3A5 (*CYP3A5*) *CYP3A5**1 allele was associated with dose reductions due to toxicity, whereas SNPs in *ABCB1* were associated with increased time-to-dose reduction. PD-related genes also appeared to play a role in toxicity risk—SNPs in vascular endothelial growth factor receptor 2 (*VEGFR2*), FMS-like tyrosine kinase 3 (*FLT3*)*,* vascular endothelial growth factor A (*VEGF-A*), and endothelial nitric oxide synthase (*eNOS*) were associated with sunitinib-related toxicities, particularly hypertension. SNPs in vascular endothelial growth factor receptor 1 (*VEGF-R1*), *VEGF-R2*, and fibroblast growth factor receptor 2 (*FGF-R2*) were also associated with PFS and/or OS. As another example, *UGT1A1**28, *UGT1A1**60, and cytochrome P450 1A2 (*CYP1A2*) rs762551 were associated with a pazopanib-related increase in bilirubin, while SNPs in hereditary hemochromatosis gene (*HFE*) were associated with increased levels of ALT. In patients receiving sorafenib, SNPs in ATP-binding cassette sub-family C member 2 (*ABCC2)* and human leukocyte antigens A (*HLA-A*) were associated with severe skin reactions, whereas *VEGF-R2* was associated with PFS and OS. Further, rs1126647 in the interleukin 28 (*IL28*) gene was associated with OS in two independent cohorts, including patients treated with sunitinib or pazopanib. Most study designs in this systematic review were candidate gene studies, focusing on CYP enzymes, CYP regulators, and drug efflux transporters, all of which seem to play a major role in various TKIs. A similar review summarizes pharmacogenes associated with impaired TKI response in patients with CML, including cytochrome P450 3A4/5 (*CYP3A4/5*), human organic cation uptake transporter 1 (*OCT1*), *ABCB1*, and *ABCG2*, as well as the effect of epigenetics and microRNAs [44].

Like TKIs in RCC and CML, epidermal growth factor receptor (EGFR) TKIs are also metabolized by cytochrome P450s and are substrates of p-glycoprotein and BCRP. Gefitinib, for example, is metabolized by *CYP3A4* and *CYP2D6*, and erlotinib is metabolized by *CYP1A2* and *CYP3A4* [45]. The gefitinib/*CYP2D6* interaction is listed in Section 1 of the FDA Table of Pharmacogenetics, stating that *CYP2D6* PMs have higher systemic concentrations and higher adverse reaction risk, but no specific dose modification is recommended [46]. Afatinib is not metabolized by cytochrome P450s and is primarily excreted through the feces, whereas osimertinib is also primarily metabolized by *CYP3A4* and dacomitinib activated by *CYP2D6*. Lastly, mobocertinib is metabolized by *CYP3A4* and *3A5*. It is well-recognized that SNPs exist in all these genes, and some evidence suggests they play a role in modulating drug exposure [45]. Further, some genes like *CYP1A2* are also affected by environmental stimuli like smoking. Cigarette smoke, in particular, produces polycyclic aromatic hydrocarbons that induce *CYP1A2* expression, which results in increased clearance of erlotinib, leading to reduced drug exposure and possibly lower efficacy [46]. Emerging drug-gene interactions identified during early-phase clinical development may allow dose stratification by genotype to optimize exposure and reduce toxicities like those observed with *CYP2C19* and tivantinib in clinical development, or *DPYD, NUDT15, TPMT*, and *UGT1A1* in the post-marketing setting.

## 6. Tyrosine Kinase Inhibitors and Pharmacodynamics to Optimize Dose/Exposure

Several examples exist in which a better understanding of drug PD (e.g., the interaction between drug-receptor-effect) may have allowed for better dosing strategies for TKIs currently on the market. For example, ibrutinib was approved at a fixed dose of 420 mg daily for the treatment of CLL. However, a study demonstrated that a dose of 2.5 mg/kg (more than half the fixed-dose in an average adult weighing 70–80 kg) resulted in 95% BTK receptor occupancy, which did not increase with further dose escalation [47]. Further, another study showed that ibrutinib 420 mg daily resulted in higher trough plasma concentrations compared with 280 mg daily but no difference in BTK receptor inhibition [48]. This is reflected in real-world data showing that nearly half of all patients require dose reductions due to adverse events or early drug discontinuation [49]. Shifting away from the traditional 3 + 3 dose escalation design to personalize pharmacology-driven dose optimization would ensure each patient (or at least most patients) gets the minimum effective and safe dose.

## 7. Monoclonal Antibodies (mAbs) and Pharmacological Methods to Optimize Dose/Exposure

Monoclonal antibodies (mAbs) target cancer cell killing via multiple mechanisms, including stimulation of the immune system by binding to antigen(s) on the surface of the cell, activating the immune system by blocking immune-checkpoint proteins, inhibiting growth factors on cancer cells, and other complex antibody-dependent cellular cytotoxicity (ADCC) mechanisms [50]. Although some mAbs have distinct E-R profiles, TDM has not been researched extensively or incorporated into clinical practice. Implementing TDM for mAbs is limited by bioanalytical challenges in quantifying mAbs in plasma. The PK for mAbs is complex, exposure-response relationships related to clinical efficacy are not typically characterized, and most have large therapeutic windows, potentially limiting the use of TDM to personalize dosing. Unlike TKIs or traditional cytotoxic chemotherapy, there are limited reports of TDM-based dosing for mAbs [51].

On the other hand, genetic predictors of toxicity and response to mAbs have been investigated and suggest potential molecular mechanisms that drive response to mAbs. For example, multiple studies have described the relationship between various Fc-gamma-receptor (*FCGR*) polymorphisms. *FCGR2A* and *FCGR3A* have been reported to affect ADCC activity. A C > T substitution (rs1801274) in the extracellular domain of *FCGR2A* and a T > G substitution (rs396991) in the extracellular domain of *FCGR3A* are coding variants that appear to alter ADCC activity. While several studies have been performed with trastuzumab and rituximab, conflicting reports have resulted in a lack of translation to the clinical setting [50]. Genetics can also impact the neonatal Fc receptor (*FcRn*) primarily through a variable number of tandem repeats (VNTR) in the promoter region [52]. For example, a study of patients receiving cetuximab showed that those harboring three repeats had reduced distribution, clearance, and increased half-life [53].

SNPs within the Programmed death-ligand 1 (*PD-L1*) gene have also been shown to affect the response to immune checkpoint inhibitors. For example, a study of patients with NSCLC receiving nivolumab showed that those with the CC and CG genotypes (rs4143815) had higher median PFS compared to patients with the GG genotype [54]. Polymorphisms in Cytotoxic T lymphocyte-associated antigen 4 (*CTLA4*) have also been associated with response to anti-CTLA4 treatment. For example, in a study of patients with metastatic melanoma receiving ipilimumab, those with rs11571316 (-577G/A) and rs3087243 (CT60G > A) homozygous genotypes had better long-term survival at 3 and 4 years [55]. Another study showed that rs4553808 (-1661G/G) was more frequent in patients with endocrine immune-related adverse events but not gastrointestinal or cutaneous adverse events [56].

PGx factors on both the tumor and host side may affect both PK and PD and subsequently play a role in modulating response to certain mAbs, particularly with immune checkpoint inhibitors. As novel biologics are developed, these molecular determinants of response and toxicity must be explored early to inform future applications of genomics-guided therapeutic management.

## 8. Conclusions

In conclusion, dose selection is an ongoing challenge in developing new anticancer agents. The design of early clinical studies based on the pharmacological characteristics of the drug is essential to ensure a successful development plan, but it is not always feasible, as it requires rapid approval following a sequential Phase 1–3 trial. In a majority of cases, experience shows that a personalized approach in dosing based on the characteristics of each patient and each drug could improve the safety and efficacy of the treatment. In clinical trials, the current approach is to follow the standard 3 + 3 design to assess DLTs. This approach is valuable and informative for the old cytotoxic agents, but it is problematic for the newer modalities such as TKIs, biologics, and ADCs. As a result, there are many cases of approved drugs requiring dose modifications later based on post-marketing studies requested by the FDA or performed by the research community. In the present review, we highlight the need for dose optimization, regulatory initiatives underway to promote dose optimization in drug development, and examples in which pharmacogenomics, pharmacokinetics, and pharmacodynamics could be used to optimize the dosing of anticancer treatments in clinical practice. A new paradigm for designing clinical trials that will precisely assess and define the optimal dose of new anticancer agents will accelerate drug approval and allow for more successful and safer drugs to enter the market.

## Figures and Tables

**Figure 1 cancers-15-03233-f001:**
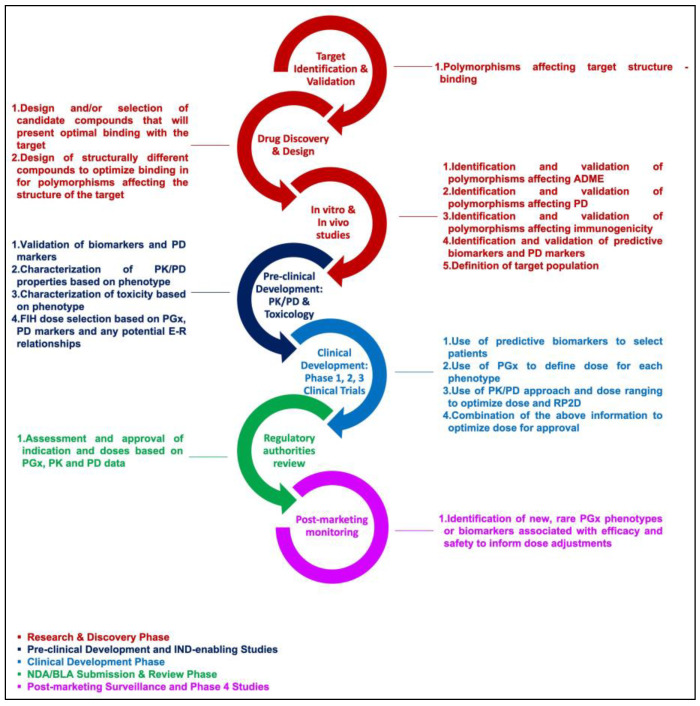
Pharmacology-informed decisions in drug development.

**Figure 2 cancers-15-03233-f002:**
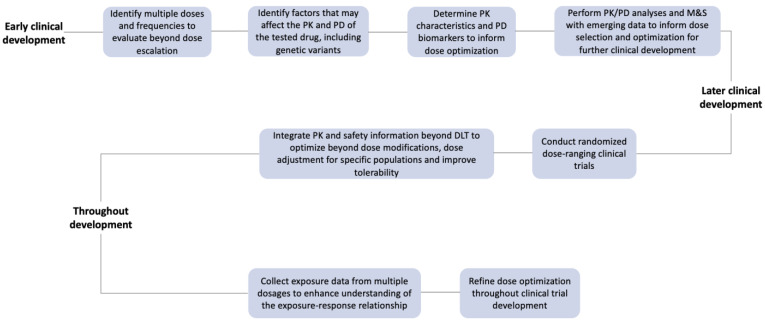
Dose optimization methods used during drug development phases.

## Data Availability

The data presented in this study are available in this article.

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
