# Peer review of "Dose Optimization in Oncology Drug Development: The Emerging Role of Pharmacogenomics, Pharmacokinetics, and Pharmacodynamics"

_cancers, 2023, doi:10.3390/cancers15123233_

Round 1
Reviewer 1 Report
The Manuscript Title “Dose Optimization in Oncology Drug Development: The 2 Emerging role of Pharmacogenomics, Pharmacokinetics and 3 Pharmacodynamics” has been written well.
Author must add role of Modeling and simulation in Dose selection. It’s a very important topic which must be added, as it is widely used in industries.
Line no. 105: Author should add detail about 3+3 method, exposure-response relationship,
Author should add flowchart for respective method of Dose optimization, in-order to use the reader understanding.
exposure-response relationship
Author Response
Dear Reviewer,
Thank you for your valuable comments, which we have considered and amended the manuscript accordingly (revised version submitted).
Below you will find the reviewer’s comments and our responses in bold.
Comments:
- The Manuscript Title “Dose Optimization in Oncology Drug Development: The 2 Emerging role of Pharmacogenomics, Pharmacokinetics and 3 Pharmacodynamics” has been written well.
We thank the reviewer for his/her comment.
- Author must add role of Modeling and simulation in Dose selection. It’s a very important topic which must be added, as it is widely used in industries.
We thank the reviewer for his/her comment. We have added a section for the role of modeling and simulations; please see Section 3 (lines 335-369).
- Line no. 105: Author should add detail about 3+3 method, exposure-response relationship,
We thank the reviewer for his/her comment. We have added details about the 3+3 method and exposure-response; please see lines 122-132.
- Author should add flowchart for respective method of Dose optimization, in-order to use the reader understanding.
We thank the reviewer for his/her comment. We added a flowchart; see Figure 2 (page 4).
Reviewer 2 Report
The review article entitled ‘Dose Optimization in Oncology Drug Development: The Emerging role of Pharmacogenomics, Pharmacokinetics and Pharmacodynamics’ Reactions' is very interesting and provides an overview of the rigorous detail that is required for dose optimization in oncology drug development. However, the author could improve the article.
· The authors could describe at what stages of drug development they could explore PG, PK, and PD need to be explored.
· What are the methods that needs to be used to determine PG, PK, PD readouts?
· Can the authors comment on other guidelines apart from FDA such as EMA and Health Canada and if any ICH guidelines that exist in this context.
· The authors mainly focused on the tyrosine kinase inhibitors, but in recent times the drug targeting in oncology has evolved considerably, the authors need to include monoclonal antibody therapy or Mabs as they are widely used to treat cancer and the PK and PD considerations for designing a dosage strategy.
· The authors could also shed some light on the immuno oncology targets such as CAR-T therapy and the latest mRNA therapeutic interventions that are being pursued for the treatment of cancer. It would be interesting to understand what are the measures that could be explored for the determining the optimal dose in these contexts.
· A schematic outline of the various stages in the process of dose optimization would be beneficial for the readers.
· References are incomplete (reference 42)
· A section for abbreviations is required (DPYD,NUDT15, TPMT,UGT1A1)
· The authors can include the article by derendorf and meibohm’ Modeling of pharmacokinetic/pharmacodynamic (PK/PD) relationships: concepts and perspectives’
Author Response
Dear Reviewer,
Thank you for your valuable comments, which we have considered and amended the manuscript accordingly (revised version submitted).
Below you will find the reviewer’s comments and our responses in bold.
Comments:
The review article entitled ‘Dose Optimization in Oncology Drug Development: The Emerging role of Pharmacogenomics, Pharmacokinetics and Pharmacodynamics’ Reactions' is very interesting and provides an overview of the rigorous detail that is required for dose optimization in oncology drug development. However, the author could improve the article.
- The authors could describe at what stages of drug development they could explore PG, PK, and PD need to be explored.
We thank the reviewer for his/her comment. A detailed description has been added to the text; please see lines 94-108.
- What are the methods that needs to be used to determine PG, PK, PD readouts?
We thank the reviewer for his/her comment. The current review focuses on describing the utility and significance of using PG, PK, and PD to optimize the dose. It is out of the scope to describe in detail the technical parts of the methods used in PG, PK, and PD analyses.
- Can the authors comment on other guidelines apart from FDA such as EMA and Health Canada and if any ICH guidelines that exist in this context.
We thank the reviewer for his/her comment. To our knowledge, EMA and Health Canada have no specific guidelines for dose optimization, only guidelines for phase 1 studies. ICH is an internationally agreed standard that ensures the ethical and scientific quality of clinical trials and does not provide any guidelines for study design, dose optimization, etc.
- The authors mainly focused on the tyrosine kinase inhibitors, but in recent times the drug targeting in oncology has evolved considerably, the authors need to include monoclonal antibody therapy or Mabs as they are widely used to treat cancer and the PK and PD considerations for designing a dosage strategy.
We thank the reviewer for his/her comment. We have added a new section for mAbs; please see section 7 (lines 536-581).
- The authors could also shed some light on the immuno oncology targets such as CAR-T therapy and the latest mRNA therapeutic interventions that are being pursued for the treatment of cancer. It would be interesting to understand what are the measures that could be explored for the determining the optimal dose in these contexts.
We thank the reviewer for his/her comment. The review focuses on dose optimization of biologics and small molecules. Other therapeutic modalities, such as cellular therapies and RNA-based therapeutics, are out of the scope of the present paper.
- A schematic outline of the various stages in the process of dose optimization would be beneficial for the readers.
We thank the reviewer for his/her comment. We added a flowchart; see Figure 2 (page 4).
- References are incomplete (reference 42)
We thank the reviewer for noticing the missing reference. We have updated the references list.
- A section for abbreviations is required (DPYD,NUDT15, TPMT,UGT1A1)
We thank the reviewer for his/her comment. The complete reference of each abbreviation is now included in the text each time a new abbreviation is mentioned for the first time. The following abbreviations are covered:
- ABCB1
- ABCC2
- ABCG2
- DPYD
- EGFR
- eNOS
- HLA-A
- IL28
- OCT1
- FGF-R2
- FLT3
- HFE
- KRAS G12C
- NR1I3
- NUDT15
- TPMT
- UGT-1A1
- VEGF-A
- VEGF-R1
- VEGF-R2
- cMET
- FCGR
- FcRn
- The authors can include the article by derendorf and meibohm’ Modeling of pharmacokinetic/pharmacodynamic (PK/PD) relationships: concepts and perspectives’
We want to thank the reviewer for identifying the paper, which we have cited in our revised manuscript. Please see reference 27 (lines 336-337).